# Seasonal and Spatial Variations of Bacterial Community Structure in the Bailang River Estuary

**Wenxun Dong** [1,2]**, Zhengguo Cui** [2,]*****, Mengjuan Zhao** [3] **and Junfeng Li** [3,]*****

[1] College of Chemistry and Molecular Engineering, Qingdao University of Science and Technology, Qingdao 266042, China; 13012410250@163.com

[2] Laboratory for Marine Fisheries Science and Food Production Processes, Laoshan Laboratory, Key Laboratory of Sustainable Development of Marine Fisheries, Ministry of Agriculture and Rural Affairs, Yellow Sea Fisheries Research Institute, Chinese Academy of Fishery Sciences, Qingdao 266071, China

[3] College of Marine Science and Biological Engineering, Qingdao University of Science and Technology, Qingdao 266042, China; zmj740587609@163.com

***** Correspondence: cuizg@ysfri.ac.cn (Z.C.); lijf1999@qust.edu.cn (J.L.)

**Abstract:** Planktonic cells are a vital part of biogeochemical nutrient cycling and play an extremely important role in maintaining the balance of water ecosystems. In this study, surface water samples were collected in three seasons (spring, summer, and winter) 10 km along the estuary of the Bailang River to assess the relationship between environmental factors and the bacterial community structure, which was determined by high-throughput sequencing. The physicochemical properties of the samples, including the pH, salinity, and inorganic nitrogen ($NH_4^+$, $NO_3^-$, and $NO_2^-$), exhibited significant seasonal variations, and the diversity and structure of the bacterial community also varied seasonally. A redundancy analysis showed that the inorganic nitrogen ($NH_4^+$, $NO_3^-$, $NO_2^-$), pH, and salinity are key factors in shaping the bacterial community composition. Among the different seasons, the core taxa of bacterial communities were the same, and *Actinobacteria*, *Cyanobacteria*, and *Proteobacteria* were the key components during the three seasons. The bacterial diversity and structure also varied seasonally, but there were no significant differences in spatial composition. Based on a phylogenetic investigation of communities by reconstruction of unobserved states analysis, nitrogen-cycle-related processes include four dominant processes: nitrogen mineralization, nitrogen fixation, dissimilatory nitrate reduction, and denitrification. These results suggest that the bacterial community structure in the waters of the Bailang River estuary is subject to seasonal rather than spatial variation. These findings provide new evidence for studies of the seasonal variation of bacterial communities in estuaries.

**Keywords:** functional genes; microbial community; nitrogen metabolism; seasonal variation; spatial variations





## 1. Introduction

In estuarine systems, planktonic microbial communities are key components in biogeochemical cycles and nutrient regeneration [1]. Planktonic organisms consume dissolved carbon nitrogen, phosphorus, and the organic matter produced by photosynthesis. This partly limits water eutrophication and promotes the biogeochemical cycle in marine ecosystems [2]. In previous studies, high-throughput 16S rDNA gene sequencing was used to study the spatial and temporal variations of bacterioplankton communities in different estuaries, such as the Pearl River Estuary [3], Columbia River coastal margin [4,5], and Baltic Sea [6]. Studies have shown that bacterial communities in water are more sensitive to environmental changes compared with those in soil [7]. Therefore, changes in a bacterial community in water can better demonstrate the relationship between seasonal changes and the diversity and structure of the bacterial community.

In the last decade, changes in the structure and composition of microbial communities in estuarine ecosystems and the factors influencing them have been researched. For example, Yu [8] found that temperature is a key factor impacting the diversity and structure of planktonic microbial communities in the surface layer of water. Yang [9] discovered that the composition of the microbial communities of Tibetan lakes go hand in hand with the change in salinity. Chi [10] observed that microbial community structure was controlled by pH, salinity, and soil organic matter. In addition, geographical distance [11] and human activity [12] can explain changes in bacterial community diversity. Previous research found that day length also affected bacterial community structure [13]. This study provides insights into microbial community patterns and the relationship between environmental factors and microbial communities.

The Bailang River, located in the south of Laizhou Bay, is one of the main rivers entering the bay. The river quality has an important impact on the water quality of Laizhou Bay. In recent years, the effects of environmental factors on the composition and diversity of the bacterial community in Laizhou Bay sediment [14] have been studied The effect of the relationship between polyaromatic hydrocarbons and the coastal bacterial community and their functional genes has been be researched, using naphthalene dioxygenase genes as biomarkers to assess the attenuation of PAHs [15]. The relationship between southern Laizhou Bay environmental factors (such as nutrients, salinity, and pH) and bacterial community structure and diversity in the water has been widely studied [16]. However, research on the Bailang Estuary has rarely been reported.

In this study, we aimed to investigate the spatial and temporal variations in the bacterial community structure of the Bailang Estuary. The potential relationship between the physicochemical properties of the water and the microbial community structure was investigated using a principal coordinates analysis (PCoA), non-metric multidimensional scaling (NMDS), and a redundancy analysis (RDA). Nitrogen metabolism functions in the bacterial community were predicted using the phylogenetic investigation of communities by reconstruction of unobserved states (PICRUSt). This study provides data support for studying the characteristics of bacterial community structure and the environmental factors affecting community shaping adjacent to the estuarine connecting Laizhou Bay.

## 2. Materials and Methods

### 2.1. Samples Collection and Analysis

Water samples from the Bailing River estuary were collected in three seasons: spring (27 May 2021), summer (16 August 2021), and winter (15 December 2020) at five sampling sites. Forty-five surface water samples (5–20 cm depth) were collected in cleaned, acid-washed polyethylene bottles. The samples were transported to the laboratory under refrigerated conditions (4 °C), and 220 mL samples of water were used for the analysis of their physicochemical properties by automated flow injection analysis; 230 mL of the same sample was filtered with a polycarbonate membrane and stored in −80 °C for DNA extraction.

### 2.2. DNA Isolation and PCR Amplification

The genomic DNA was extracted with a DNA Isolation Kit (MoBio, Carlsbad, CA, USA). Each sample was extracted in triplicates. The V3-V4 region of the 16S rDNA gene was collected from the bacterial DNA by polymerase chain reaction (PCR) using primers 338F (5′-ACTCCTACGGGAGGCAGCAG-3′) and 806R (5′-GGACTACHVGGGTWTCTAAT-3′).

### 2.3. Statistical Analysis

High-throughput sequencing was performed using the Quantitative Insights Into Microbial Ecology 2 (QIIME2) (Personalbio Shanghai, Shanghai, China) [17]. Statistical analysis was carried out using the QIIME2, including the alpha diversity indices OTUs, Chao1, Shannon (H), Pielou's evenness (J), and Good's coverage; beta diversity indices included the PCoA, hierarchical cluster analysis, NMDS, and RDA. Pearson's correlation

test was conducted using Statistical Product and Service Solutions (SPSS) (v2.15.3) [18]. Functional genes were predicted by PICRUSt. The Kyoto Encyclopedia of Genes and Genomes (KEGG) was used to predict the cellular metabolic activity.

## 3. Results and Discussion

### 3.1. Physicochemical Properties of the Samples

The properties of each sampling site are shown in (Table 1). The pH of the samples gradually decreased from spring to winter, which may be related to plant photosynthesis. Photosynthesis was strong in the spring, resulting in a lower carbon dioxide content in the water. The pH of the samples gradually decreased from spring to winter, which may be related to plant photosynthesis. The surface salinity gradually decreased from spring to winter, likely due to the lower precipitation and river runoff in the spring, resulting in seawater intrusion and leading to the high salinity in the spring. A Pearson correlation analysis showed that $NH_4^+$-N had a negative correlation with salinity and pH ($p < 0.05$, Table 2), and T ($p < 0.01$, Table 2) was positively correlated with $NO_3^-$-N and $NO_2^-$-N.

**Table 1.** Physicochemical properties of all samples in the Bailang estuary.

| Sites | Season | pH | Salinity (psu) | T/°C | $NH_4^+$-N (mg/L) | $NO_3^-$-N (mg/L) | $NO_2^-$-N (mg/L) |
|---|---|---|---|---|---|---|---|
| MTL1 | Spring | 8.71 | 14.22 | 24.0 | 0.0167 | 0.0655 | 0.0065 |
| | Summer | 8.16 | 13.94 | 28.1 | 0.4962 | 0.0775 | 0.0042 |
| | Winter | 7.94 | 9.82 | 5.0 | 0.2958 | 0.0404 | 0.0021 |
| HLH2 | Spring | 8.69 | 18.87 | 25.7 | 0.0241 | 0.0562 | 0.0057 |
| | Summer | 8.33 | 13.01 | 27.9 | 0.2783 | 0.0546 | 0.0036 |
| | Winter | 8.26 | 10.36 | 3.2 | 0.0558 | 0.0295 | 0.0018 |
| ZLM3 | Spring | 8.59 | 16.67 | 24.5 | 0.0162 | 0.0325 | 0.0014 |
| | Summer | 8.31 | 13.89 | 28.1 | 0.0824 | 0.0751 | 0.0070 |
| | Winter | 8.15 | 10.56 | 2.4 | 0.0650 | 0.0310 | 0.0017 |
| XD4 | Spring | 8.61 | 17.52 | 22.0 | 0.0096 | 0.0348 | 0.0012 |
| | Summer | 8.28 | 16.81 | 27.9 | 0.0617 | 0.0938 | 0.0088 |
| | Winter | 8.21 | 11.71 | 3.0 | 0.0807 | 0.0377 | 0.0024 |
| YTMT5 | Spring | 8.69 | 17.81 | 22.0 | 0.0039 | 0.0398 | 0.0022 |
| | Summer | 8.35 | 16.94 | 28.8 | 0.0163 | 0.0635 | 0.0069 |
| | Winter | 8.23 | 11.92 | 3.7 | 0.0709 | 0.0306 | 0.0016 |

**Table 2.** Pearson correlation coefficient matrix between nutrients and other water properties.

| | pH | S | T | $NH_4^+$-N | $NO_3^-$-N | $NO_2^-$-N |
|---|---|---|---|---|---|---|
| pH | 1 | | | | | |
| S | 0.740 ** | 1 | | | | |
| T | 0.517 * | 0.478 | 1 | | | |
| $NH_4^+$-N | −0.608 * | −0.717 ** | 0.047 | 1 | | |
| $NO_3^-$-N | 0.002 | 0.106 | 0.711 ** | 0.259 | 1 | |
| $NO_2^-$-N | 0.126 | 0.289 | 0.644 ** | −0.062 | 0.917 ** | 1 |

* $p < 0.05$, significant correlation. ** $p < 0.01$ level of significant correlation.

### 3.2. Sequencing Statistics and Diversity of the Bacterial Community

Following the high-throughput sequencing, a total of 3,063,038 clean sequences were generated from 45 samples, and the OTU numbers ranged from 1052 to 3614. A coverage value of 0.98 confirmed that the microbial community might have been correctly represented by the OTUs (Table 3).

**Table 3.** Richness and α-diversity of the season samples.

| Samples | Chao1 | PD Whole Tree | Coverage | Observed Species | Pielou e | Shannon | Simpson |
|---------|-------|---------------|----------|------------------|----------|---------|---------|
| Spring | 2036.40 | 129.74 | 98.84% | 1657.47 | 0.64 | 6.83 | 96.16% |
| Summer | 2439.87 | 123.49 | 98.63% | 1982.00 | 0.64 | 7.00 | 95.84% |
| Winter | 2521.18 | 144.62 | 98.67% | 2109.88 | 0.65 | 7.12 | 95.41% |

Chao1 confirmed that the taxon richness increased from spring to winter. Differences in the Shannon values between samples in different seasons were determined using a *t*-test (Table 3). The Shannon value showed an upward trend from spring to winter, which was the same conclusion as the Chao1 values; however, there was no significant difference between the different seasonal samples ($p > 0.05$). The observed species demonstrated that bacterial diversity increased from spring to winter ($p < 0.01$).

A PCoA of the Bray–Curtis distance was used to evaluate the spatial and temporal distribution of the bacterial communities in the water samples. As shown in Figure 1, the samples were divided into three clusters and displayed seasonal changes ($p < 0.01$). Figure 2 shows the differences in the bacterial communities at the OTU level. These results were similar to those of the PCoA (Figure 1). The samples were divided into three clusters according to season. An osim analysis was used to assess differences in the bacterial communities across seasons, indicating that the bacterial communities were significantly different in different seasons ($p < 0.001$), but there was little variation between the sampling sites. The bacterial communities in different seasons were analyzed using a nonparametric multivariate analysis of variance (NPMANOVA) (Supplementary Table S1). The results showed that the bacterial community varied significantly in the spring ($p < 0.01$), summer ($p < 0.01$), and winter ($p < 0.05$).

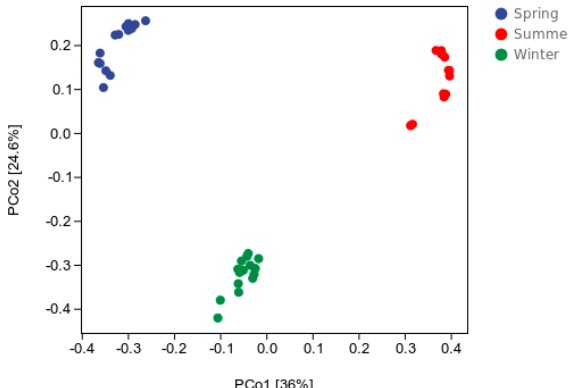

**Figure 1.** Principal coordinates analysis (PCoA).

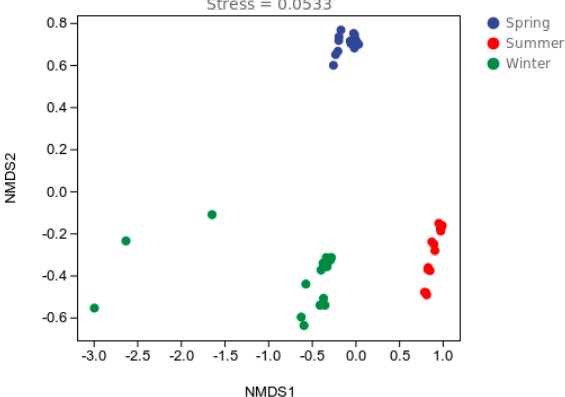

**Figure 2.** Non-metric multidimensional scaling (NMDS).

To study the influence of the water's physicochemical properties on the bacterial community structure of the Bailang River estuary, we analyzed the contribution of environmental factors to the variance in the bacterial community structure in phylum using an RDA (Figure 3a). The RDA1 axis explains 33.24% of the total variance, whereas the RDA2 axis interprets 14.62%. From Figure 3a, the samples are distributed in different quadrants according to the season. The spring samples were mainly gathered in Quadrant 3, whereas the summer samples were gathered in Quadrants 1 and 4. The winter samples were primarily gathered in Quadrant 2. At the phylum level, the RDA analysis revealed that except for the pH ($p > 0.05$), the other factors, S (F = 0.326, $p < 0.01$), T (F = 0.596, $p < 0.01$), $NH_4^+$ (F = 0.283, $p < 0.01$), $NO_3^-$ (F = 0.230, $p < 0.01$), and $NO_2^-$ (F = 0.264, $p < 0.01$), were significant factors affecting bacterial community diversity. This result showed that the bacterial community composition and diversity in water are closely related to the physicochemical properties of the water (Figure 3a). Previous studies have confirmed that natural salinity gradients, pH [19], DO, and even geographic distance [11] could structure the bacterial community composition and diversity in estuarine systems. Salinity is one of the main factors influencing the distribution of microbial communities, and it has a strong influence on soil [20], river sediments [21], and surface and bottom waters [2]. The above results showed that salinity was also an important factor in the bacterial community assembly in the Bailang River estuary. Temperature is also one of the main factors influencing the distribution of microbial communities. Studies have shown that bacterial communities in water are more temperature−sensitive than soil communities [22]. In addition to physical factors, the concentrations of dissolved inorganic nitrogen, such as $NH_4^+$, $NO_3^-$, and $NO_2^-$, are key regulators of the structure and composition of the microbial community. $NH_4^+$ has been reported to be an important factor affecting community composition [23]. The network analysis showed a correlation between the species and seasonal variables (Figure 3b). The results suggest that in the Bailang estuary, at a species level, bacterioplankton showed significant variations during different seasons.

### 3.3. Bacterial Community Composition and Structure at the Phylum and Genus Level

In this study, the bacterial community composition and structure in different seasons in the Bailang River estuary were analyzed based on the phylum−level and genus−level bacterial relative abundance. Figure 4a showed that *Actinobacteria*, *Cyanobacteria*, and *Proteobacteria* were the major components. *Actinobacteria* was the most abundant bacterial phylum in all the water samples, followed by *Cyanobacteria*, *Proteobacteria*, and *Bacteroidetes*. *Actinobacteria* gently increased in relative abundance from spring (42.86%) to summer (46.13%) and then declined sharply in the winter (27.15%). *Actinobacteria* was the core composition in the bacterial phylum in all samples. It is capable of decomposing organic matter into the inorganic matter; therefore, the molecules can be reused by microbes, which are usually more active in oligotrophic environments [24]. Previous studies have shown that the relative abundance of actinomycetes decreases with an increasing nutrient concentration [25]. According to the Pearson correlation analysis, *Actinobacteria* was positively correlated with T (Supplementary Table S2), which demonstrated that environmental factors influenced the bacterial community composition.

*Proteobacteria* showed an opposite seasonal trend to *Actinobacteria*: its relative abundance decreased from spring (11.38%) to summer (6.19%) and increased in the winter (26.26%). In *Proteobacteria*, the seasonal average relative abundance at the class level showed that *Alphaproteobacteria* (7.87%, 3.39%, and 15.13%) and *Gammaproteobacteria* (2.78%, 1.99%, and 10.32%) were the major components. *Alphaproteobacteria* and *Gammaproteobacteria* showed the same trend, decreasing from spring to the summer and then increasing in the winter. *Proteobacteria* (*Alpha*, *Gamma*, and *Delta*) have diverse metabolic pathways that are critical for nutrient cycling, conversion, and the remineralization of substances such as organic carbon and nitrogen [26–28]. *Alphaproteobacteria* is generally abundant in marine waters [29,30]. Figure 4b shows relatively small differences in the abundance of *Alphaproteobacteria* across the seasons. Previous studies have shown that *Gammaproteobacteria* plays

an important role in denitrification and phosphorus removal [31,32]. The results of the Pearson correlation analysis show that *Alphaproteobacteria* exhibited a negative correlation with T, $NO_3^-$, and $NO_2^-$ ($p < 0.05$), *Gammaproteobacteria* was only negatively correlated with T ($p < 0.05$, Supplementary Table S3).

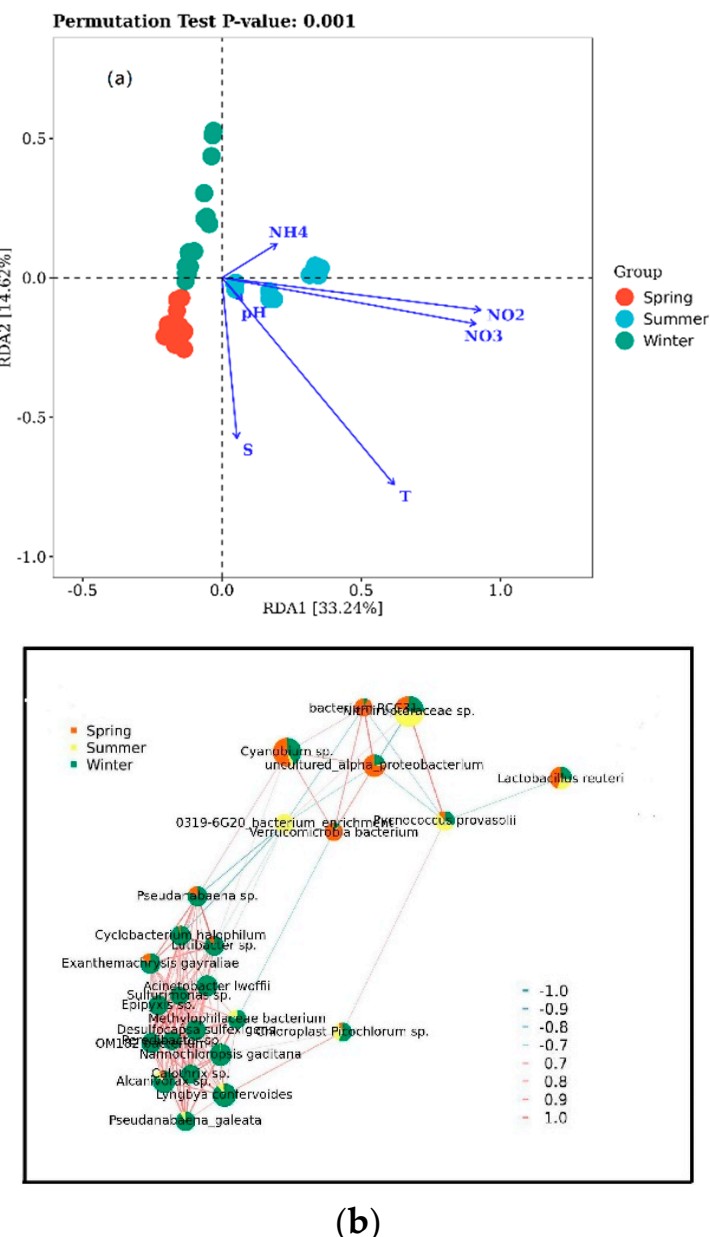

**Figure 3.** Redundancy analysis (RDA) shows the relationship between physico−chemical properties and microbial community structures at the phylum level (**a**); correlation network among species and seasonal variables in the Bailang estuary (**b**).

*Cyanobacteria* gradually decreased from spring (36.97%) to summer (34.87%) and then sharply decreased in the winter (23.43%). *Bacteroidetes* gradually increased in relative abundance from spring (4.54%) to winter (9.35%). In general, *Cyanobacteria* are responsible for the majority of the primary producers of bacteria in aquatic environments [33,34]. Therefore, they are of paramount ecological importance as primary producers in the global food chain [35]. *Cyanobacteria* were positively associated with T, $NO_3^-$, and $NO_2^-$ ($p < 0.05$) but negatively correlated with *Proteobacteria* ($p < 0.05$, Supplementary Table S2).

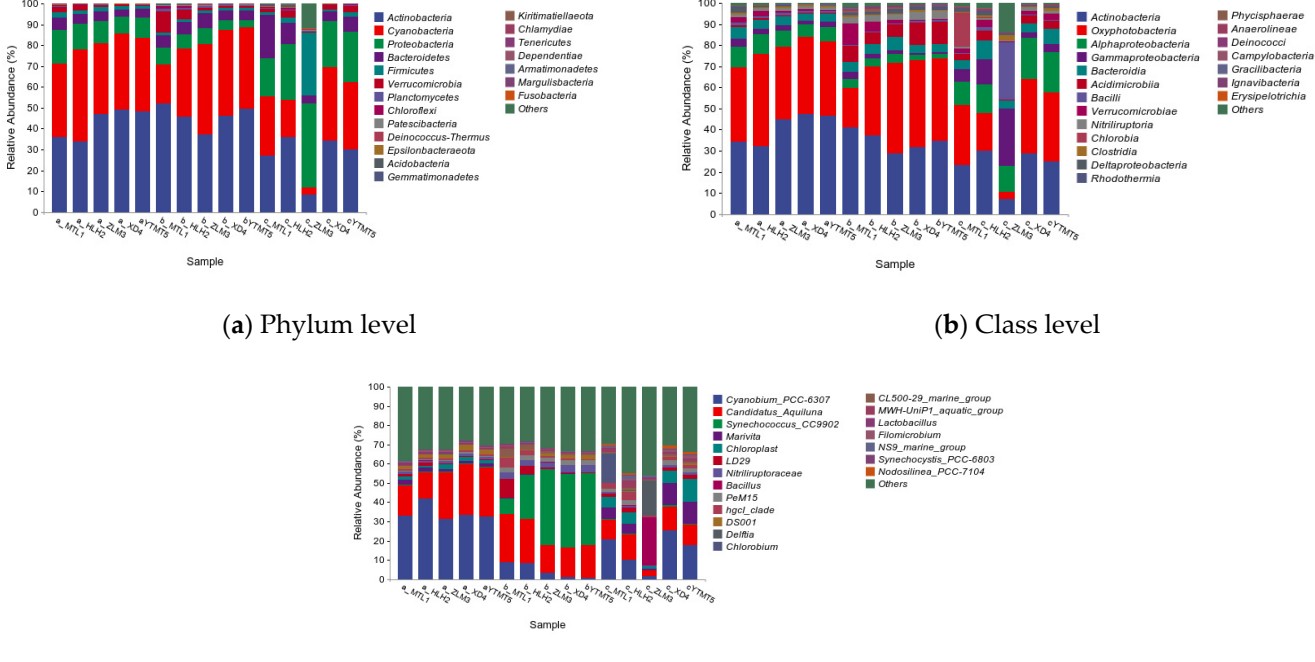

**Figure 4.** Taxonomic classification of the bacterial community at (**a**) phylum, (**b**) class levels, and genus levels (**c**); a, b, and c indicate the spring, summer, winter. MTL1, HLH2, ZLM3, XD4, and YTMT5 indicate the sampling locations in the Bailang Estuary. e.g., "a_MTL1" mean sample of site MTL1 in spring.

*Verrucomicrobia* (1.51–3.54%), *Planctomycetes* (0.18–0.80%), *Chloroflexi* (0.05–0.52%), *Patescibacteria* (0.24–0.38%), *Deinococcus-Thermus* (0.02–0.29%), *Nitriliruptoraceae* (0.95–3.26%) are present with slight differences in relative abundance in all samples. Most of these can be found in diverse environments, such as the soils, sediments, and waters in coastal areas [36]. *Firmicutes* are more adaptable to lower temperatures than other bacteria and can degrade organic nitrogen, indicating that the nitrogen cycle in this region may be active in the winter [14]. *Planctomycetes* have been detected in various environmental and wastewater treatment processes; they are important bacteria that play key roles in the anaerobic ammonium oxidation (anammox) process [37]. *Nitriliruptoraceae* perform significant functions in suppressing harmful organisms [38].

At the genus level, *Cyanobium* sp. PCC-6307, *Candidatus Aquiluna*, and *Synechococcus* sp. CC9902 were the three dominant taxa in these samples. However, they exhibited vastly different seasonal variations. *Cyanobium* sp. PCC-6307 gradually decreased from spring (34.36%) to summer (4.55%) and then sharply increased in the winter (15.02%). Conversely, *Synechococcus* sp. CC9902 sharply increased from spring (0.22%) to summer (29.12%), becoming the core genus, but then sharply decreased in the winter (0.40%). The Pearson analysis result showed that *Cyanobium* sp. PCC-6307 was positively correlated with pH ($p < 0.05$, Supplementary Table S4).

*Candidatus Aquiluna* belongs to *Actinobacteria*. The seasonal relative abundances of *Candidatus Aquiluna* gradually decreased from spring (21.18%) to summer (18.89%) and then sharply decreased in the winter (9.89%). Pearson analysis showed that *Candidatus aquiluna* was positively correlated with T ($p < 0.05$, Supplementary Table S4). In spring, *LD29 PeM15* and *hgcIclade* accounted for <1% of the total sequences, but these genera subsequently increased to above 1.0% in the summer or winter (Figure 4c). *LD29* is one of the most abundant species in coastal and brackish environments [39]. However, in spring, *Marivita,* not only can degrades hydrocarbons [40] but also denitrify under aerobic conditions [41], accounts for >1.0% of the total sequences but decreases to below 0.5% in the summer. A small amount of *Nitrospirae*, a key player in the nitrification process, was

detected in the summer (<0.002%) and winter (>0.013%). Numerous genera showed a positively or negatively correlation with T, $NO_3^-$-N, and $NO_2^-$-N ($p < 0.05$, Supplementary Table S4). This finding further confirms that environmental factors may also shape bacterial community structure to some extent.

### 3.4. Prediction of Nitrogen Metabolism Function in the Bacterial Community in the Bailang River Estuary

The functions of the water's bacterial community were predicted using PICRUSt, and physical and chemical factors may cause changes in the functional potential of the community. As nitrogen is a highly variable nutrient and an important indicator of water quality, the functional genes and pathways of nitrogen metabolism in nature are crucial to environmental remediation. The results showed that most genes significantly differed among the seasons. Numerous nitrogen metabolism functional genes have clearly negative or positive correlations with T, pH, salinity, and inorganic nitrogen ($NH_4^+$, $NO_3^-$, and $NO_2^-$) (Supplementary Table S5).

Figure 5 shows the KEGG-predicted genes for nitrogen cycling in different seasons. The results showed that the low abundance of ammonia monooxygenase genes in the estuary led to a weaker nitrification process in the estuarine ecosystem [EC:1.14.99.39]. Therefore, it is speculated that the main processes of nitrogen cycling in estuaries are nitrogen mineralization (NM), nitrogen fixation (NF), dissimilatory nitrate reduction (DNRA), and denitrification (ND).

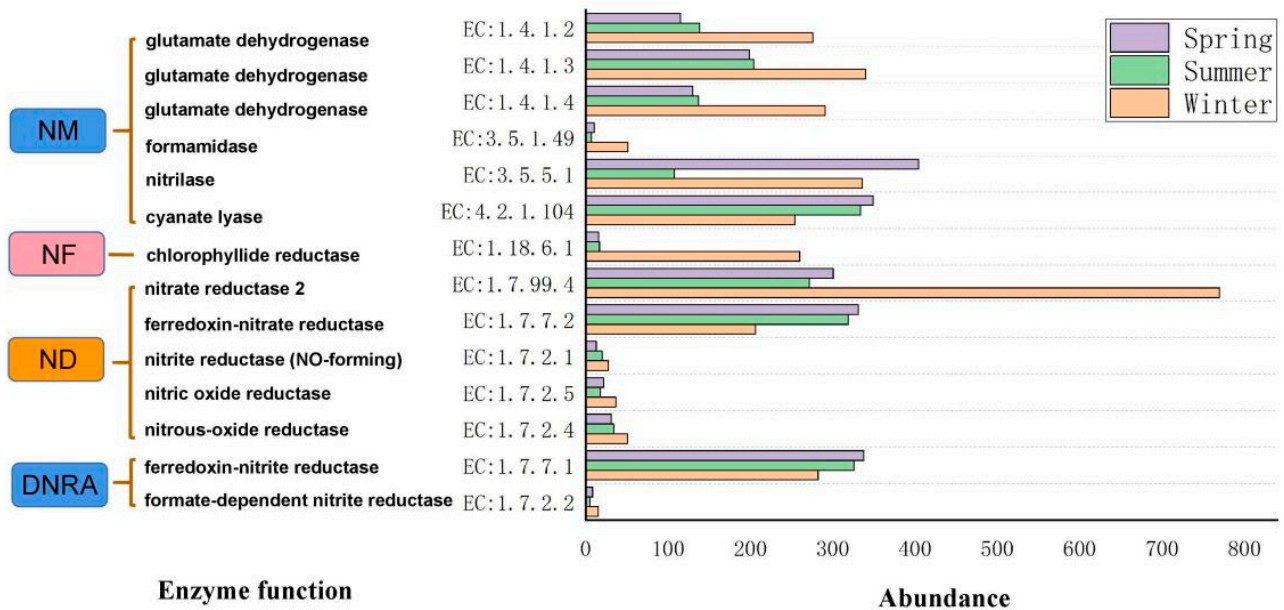

**Figure 5.** Functional genes related to nitrogen metabolism in samples from different seasons.

Previous research has shown that *Bacillus*, *Pseudomonas*, *Rhodobacter*, *Flavobacterium*, and *Dechloromonas* contribute substantially to organic nitrogen mineralization rates. The gene abundance of nitrilase [EC:3.5.5.1] decreased and reached a minimum level in the summer and the maximum level in the winter. For formamidase [EC:3.5.1.49], it could be observed in a different season. The gene abundance of cyanate lyase [EC:4.2.1.104] was positively correlated with pH, salinity, and T (Supplementary Table S5), denoting that cyanate-degrading bacteria are more suitable for growth in the summer. The genes abundance of glutamate dehydrogenase ([EC:1.4.1.3] and [EC:1.4.1.4]) in the NM process were negatively correlated with T and salinity (Supplementary Table S5), which is consistent with the findings of Walpola and Arunakumara [42]. The NM process was weak in spring and summer, which may be related to plant competition for nutrients between plants and microorganisms.

NF functional bacteria included *Proteobacteria*, *Firmicutes*, *Actinomycetes*, *Verrucomicrobia*, *Cyanobacteria*, and *Spirochaetes*. The expression of NF genes abundance was significantly higher in winter than in the other seasons. The photosynthetic [EC:1.18.6.1] gene abundance of the nitrogen-fixing process was negatively correlated with the pH and T (Supplementary Table S5). The NF process is a significant source of nitrogen entering the biosphere [43,44]. Previous studies have reported that the relationship between nitrogen-fixing bacteria and temperature is constantly changing, being negatively correlated with nitrogen-fixing bacteria within ranges of 5–10 °C and 20–30 °C and being positively correlated with nitrogen-fixing bacteria within 10–20 °C [45]. The results of Brauer indicated that as the temperature increases [46], the sulfate-reducing bacteria would accelerate the dissimilatory reduction of sulfate to sulfide, which would inhibit the activity of nitrogen fixation bacteria and then reduce the nitrogen fixation rate to some extent [45].

In this study, a higher abundance of ND function was found in the winter, probably because of higher *Bacillus* abundance. The ND-related genes [EC:1.7.99.4], [EC:1.7.2.5], and [EC:1.7.2.4] were all negatively correlated with T (Supplementary Table S5). The relative abundance of denitrification genes [EC:1.7.99.4, EC:1.7.2.5] was positively correlated with T, which may be related to temperature or may be affected by the nitrate-nitrogen content (Table 1). Strangely, at the ZLM3 site, nitrate reductase [EC:1.7.99.4] was significantly higher than at other sites in the spring (Supplementary Table S6). The high sediment content of the water may cause this. Previous studies have shown that denitrification rates are closely related to salinity gradients and decrease with increasing salinity [47]. The salinity may cause physiological stress to denitrification and thus affect the denitrification rate [48]. However, no significant correlation between salinity and denitrification was found in our study. The study also found that the denitrification genes [EC:1.7.2.4], [EC:1.7.2.5], [EC:1.7.2.1], and [EC:1.7.99.4] demonstrated higher abundances in the winter than in the other seasons because ND is more favored at lower temperatures [49].

*Proteobacteria*, *Chloroflexi*, *Verrucomicrobia*, *Acidobacteria*, and *Bacteroidetes* were the main DNRA functional bacteria. *Proteobacteria* and *Bacteroidetes* was were positively correlated with the DNRA processes. The Pearson correlation analysis showed that the DNRA-related genes [EC:1.7.2.2] were negatively correlated with T and salinity, respectively (Supplementary Table S5). DNRA is a biological process that retains nitrogen in the ecosystem in its bioavailable form, $NH_4^+$ [50,51]. This is an important pathway for nitrate transformation in aquatic environments. In this study, the abundance of the DNRA process-related enzyme gene [EC:1.7.7.1] was relatively higher in the spring, followed by the summer and winter. This may be because the temperature (22–25 °C) in the spring is more suitable for the growth of DNRA bacteria. Previous studies have shown that DNRA bacteria are more efficient in reducing nitrate at temperatures of 14–17 °C [52]. There is a competitive relationship between ND and DNRA in which the efficiency is mainly controlled by temperature [49]. Changes in nitrogen-metabolism-related functions and pathways further illustrated crucial biogeochemical processes and elemental cycling in estuaries. Many studies point out that as the DNRA process increases with an increasing salinity, the decreased salinity of the DNRA process decreases, and the ND process increases with decreasing salinity [53].

## 4. Conclusions

The high-throughput sequencing and PICRUSt analyses revealed the relationship between bacterial community structure and diversity with seasonal variations and potential metabolic functions in the Bailang River estuary. The water properties varied significantly with the season, and pH, $NH_4^+$, and salinity significantly affected bacterial community aggregation in the estuary region. Bacterial community structure and diversity were shaped by seasonal changes at the phyla and genus level rather than the spatial level. T, pH, salinity, and nutrients were the key factors affecting the bacterial community composition. *Actinobacteria*, *Cyanobacteria*, and *Proteobacteria* were the predominant phyla in water, and *Cyanobium* sp. PCC-6307, *Candidatus aquiluna* and *Synechococcus* sp. CC9902 were the

most abundant genera, which were greatly affected by the seasonal variation. A PICRUSt analysis showed that the abundance of NM, NF, denitrification, and DNRA-related genes in the samples was relatively higher, and we can speculate that these processes were the dominate nitrogen cycling processes in the estuary. Generally, we understood the changes in bacterial community structure and function under changes in season, salinity, and nutrient stress. This study provides new evidence for the seasonal variation of bacterial community structure and diversity in Bailang River estuary and provides a reference for the environmental protection of the estuary.

**Supplementary Materials:** The following supporting information can be downloaded at: https://www.mdpi.com/article/10.3390/jmse11040825/s1, Table S1: Adonis analysis; Table S2: Pearson correlation analysis of Phylum level; Table S3: Pearson correlation analysis of Class level; Table S4: Pearson correlation analysis of Genus level; Table S5: Pearson correlation analysis of nitrogen cycle function genes; Table S6: Different site of nitrogen cycle function genes.

**Author Contributions:** Conceptualization, Z.C. and J.L.; methodology, Z.C. and J.L.; software, M.Z.; formal analysis, M.Z.; investigation, W.D. and M.Z.; resources, W.D.; writing—original draft preparation, W.D.; and writing—review and editing, Z.C. and J.L. funding acquisition, Z.C. and J.L. All authors have read and agreed to the published version of the manuscript.

**Funding:** Ministry of Agriculture and Rural Affairs Financial Project "Fisheries Assessment in the Yellow Sea & Bohai Sea", Basic Scientific Research Project of Chinese Academy of Fishery Sciences (2021XT0604), and Major Scientific and Technological Innovation Project (MSTIP) of Shandong (2021CXGC010705, 2019JZZY020705).

**Institutional Review Board Statement:** Not applicable.

**Informed Consent Statement:** Not applicable.

**Data Availability Statement:** The dataset presented in this study is available on valid request from the corresponding authors.

**Acknowledgments:** The article would never have existed without the guide and modification of Junfeng Li and Zhengguo Cui. Thanks for their help in my studies and life over the past three years.

**Conflicts of Interest:** The authors declare no conflict of interest.

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
