# Peer review of "Seasonal and Spatial Variations of Bacterial Community Structure in the Bailang River Estuary"

_jmse, doi:10.3390/jmse11040825_

Round 1

Reviewer 1 Report

Dear authors,

This is a great contribution about marine science micro biome and I have some point to improve before acceptance.

1. I recommend to analysis significant differences between season by NPMANOVA analysis 

2. Please construct relative network between species in each season and then discover more about this .

regards,

Author Response

Manuscript No.: jmse-2325703

Manuscript Title: Seasonal and spatial variations of bacterial community structure in the Bailang River estuary

Dear Prof.

We would like to thank you for the valuable comments and suggestions on our manuscript (jmse-2325703). According to the constructive comments and suggestions, we have revised our manuscript carefully. All changes have been emphasized in the highlighted revision in red in the manuscript. We hope that these revisions are satisfactory and the revised manuscript can be accepted for publication in Journal of Marine Science and Engineering. Detailed lists of revisions as well as our point-to-point response to your comments are listed below.

Sincerely yours

Responses to the reviewer

This is a great contribution about marine science microbiome and I have some point to improve before acceptance.

Response : We greatly thank you the positive comments.

Advice 1. I recommend to analysis significant differences between season by NPMANOVA analysis.

Answer: Nonparametric multivariate analysis of variance (NPMANOVA) is also named Permutational multivariate analysis of variance (PERMANOVA) or Adonis.We have changed “multivariate analysis of variance (Adonis) analysis” to “Nonparametric multivariate analysis of variance (NPMANOVA)”. Please see line 144-145.

Advice 2. Please construct relative network between species in each season and then discover more about this.

Answer:Thank you for your advice about the manuscript, those will help us greatly to improve the quality of our articles.

We have added correlation network analysis between species and seasonal variables in line 175-178 as “The network analysis showed correlation between species and seasonal variables (Figure 3b). The results support that bacterioplankton at species lever in the Bailang estuary were significant variations in different seasons.”. Correlation network has added, as Figure 3b in line 148.

We have submitted manuscript, Please see the attachment

Reviewer 2 Report

The studies concern the distribution of microorganisms at the Bailang River estuary and their correlation with environmental factors. The data help to assess not only the diversity of taxa depending on the season, but also to draw a conclusion about their metabolic activity and contribution to nitrogen-cycle-related processes. The work has been done well, the data is presented quite clearly. However, the manuscript needs a revision before publication.

Please consider my suggestions as listed below.

Comments

In the introduction, I recommend referring also, for example, to the work of Yu at al. (Shi Yu, Ruoxue He, Ang Song, Yadan Huang, Zhenjiang Jin, Yueming Liang, Qiang Li, Xiaohong Wang, Werner E. G. Müller, Jianhua Cao Spatial and temporal dynamics of bacterioplankton community composition in a subtropical dammed karst river of southwestern China. MicrobiologyOpen. 2019;8:e849.https://doi.org/10.1002/mbo3.849), the authors raise similar issues in their research.

Please follow exactly the instructions for Authors when preparing a manuscript.

In the text, reference numbers should be placed in square brackets [ ], and placed before the punctuation; for example [1], [1–3] or [1,3].

I recommend that the abbreviation be introduced not in the Abstract, but in the text of the manuscript itself at the place of their first mention.

Be careful with abbreviations, do not forget to decode them at the first mention.

Carefully check the text for typos: italics for Latin taxa designations, capital letters after the comma, absence of spaces between words, etc.

I recommend reading the work - Oren A, Garrity GM. Valid publication of the names of forty-two phyla of prokaryotes. Int J Syst Evol Microbiol 2021; 71:5056- in connection with the reclassification of a number of large taxa

Make the Back Matter in accordance with the recommendations of the publisher and in the following order:

Supplementary Materials

Author Contributions: For research articles with several authors, a short paragraph specifying their individual contributions must be provided. The following statements should be used “Conceptualization, X.X. and Y.Y.; methodology, X.X.; software, X.X.; validation, X.X., Y.Y. and Z.Z.; formal analysis, X.X.; investigation, X.X.; resources, X.X.; data curation, X.X.; writing—original draft preparation, X.X.; writing—review and editing, X.X.; visualization, X.X.; supervision, X.X.; project administration, X.X.; funding acquisition, Y.Y. All authors have read and agreed to the published version of the manuscript.” Please turn to the CRediT taxonomy for the term explanation. Authorship must be limited to those who have contributed substantially to the work reported.

Acknowledgment

The references part must be written according to the requirements of the journal, where the year of publication is put in its correct place and with the correct italics and bold font style.

Minor comments

Line 111 “… gradually decreased from spring to winter (7.94-8.71),…” The values in brackets correspond to only one sample out of five, I recommend removing these values from the text, it is clear from the table that there is a gradual decrease in pH.

Line 113 “…content in the waters; the higher the pH, the gradually decreased…” it is unclear why to refer to the pH value?

Figure 3 Add the quadrant designations that are in the text

Figure 4 Specify what the indices a, b, c of the sample sites mean, e.g., "a_MTL1"

Line 213 italics for Alphaproteobacteria

Line 215-217 “… but Gammaproteobacteria was only correlated with T” negative or positive correlation?

Line 238-239 “…Cyanobium_PCC-6307, Candidatus_Aquiluna and Synechococcus_CC9902…” incorrect designation of taxa, I believe it means, Cyanobium sp. PCC-6307, Candidatus Aquiluna and Synechococcus sp. CC9902.

Author Response

Manuscript No.: jmse-2325703

Manuscript Title: Seasonal and spatial variations of bacterial community structure in the Bailang River estuary

Dear Prof.

We would like to thank you for the valuable comments and suggestions on our manuscript (jmse-2325703). According to the constructive comments and suggestions, we have revised our manuscript carefully. All changes have been emphasized in the highlighted revision in red in the manuscript. We hope that these revisions are satisfactory and the revised manuscript can be accepted for publication in Journal of Marine Science and Engineering. Detailed lists of revisions as well as our point-to-point response to your comments are listed below.

Sincerely yours

Responses to the reviewer 2

The studies concern the distribution of microorganisms at the Bailang River estuary and their correlation with environmental factors. The data help to assess not only the diversity of taxa depending on the season, but also to draw a conclusion about their metabolic activity and contribution to nitrogen-cycle-related processes. The work has been done well, the data is presented quite clearly. However, the manuscript needs a revision before publication.

Response: We greatly thank you the positive comments.

Advice 1. In the introduction, I recommend referring also, for example, to the work of Yu at al. (Shi Yu, Ruoxue He, Ang Song, Yadan Huang, Zhenjiang Jin, Yueming Liang, Qiang Li, Xiaohong Wang, Werner E. G. Müller, Jianhua Cao. Spatial and temporal dynamics of bacterioplankton community composition in a subtropical dammed karst river of southwestern China. Microbiology Open.), the authors raise similar issues in their research.

Answer: Thank you for your advices about the manuscript, those will help us greatly to improve the quality of our articles.

We have cited this paper in introduction. Please see line 50-51

“Yu [8] found that temperature is a key factor that impact the planktonic microbial communities diversity and structure in the surface layer of water.

Advice 2. Please follow exactly the instructions for Authors when preparing a manuscript.

In the text, reference numbers should be placed in square brackets [ ], and placed before the punctuation; for example [1], [1–3] or [1,3].

Answer: We have revised the format of the article according to the author instructions.

line 38: changed ‘(Cotner et al., 2002)’ to ‘[1]’.

line 40: changed ‘(Griffiths et al.,2017)’ to ‘[2]’.

line 43: changed ‘(Li et al., 2017a)’ to ‘[3]’.

line 43: changed ‘(Fortunato et al., 2012; Fortunato et al.,2013)’ to ‘[4,5]’.

line 44: changed ‘(Herlemann et al., 2011)’ to ‘[6]’.

line 45: changed ‘(Kritzberg et al., 2022)’ to ‘[7]’.

line 50: changed ‘(Sunagawa et al.,2015)’ to ‘[8]’.

line 52: changed ‘(Yang et al., 2016)’ to ‘[9]’.

line 53: changed ‘(Chi et al.,2021)’ to ‘[10]’.

line 55: changed ‘( Dai et al., 2017)’ to ‘[11]’.

line 55: changed ‘(Simonato et al., 2010)’ to ‘[12]’.

line 57: changed ‘(Gilbert et al., 2011)’ to ‘[13]’.

line 62: changed ‘(Wang et al., 2014)’ to ‘[14]’.

line 64: changed ‘(Xu et al., 2019)’ to ‘[15]’.

line 66: changed ‘(Yang et al., 2022)’ to ‘[16]’.

line 95: changed ‘(Li et al., 2018b)’ to ‘[17]’.

line 99: changed ‘(Li et al., 2019a)’ to ‘[18]’.

line 165: changed ‘(Kirs et al., 2017)’ to ‘[19]’.

line 165: changed ‘(Dai et al., 2017)’ to ‘[11]’

line 168: changed ‘(Hu et al., 2014)’ to ‘[20]’

line 168: changed ‘(Ibekwe et al., 2016)’ to ‘[21]’

line 168: changed ‘(Liu et al., 2015)’ to ‘[2]’

line 172: changed ‘(Pomeroy et al., 2001)’ to ‘[22]’

line 175: changed ‘(Li et al.,2019b)’ to ‘[23]’

line 189: changed ‘(Wang et al., 2020b)’ to ‘[24]’

line 191: changed ‘(Wang et al., 2016)’ to ‘[25]’

line 207: changed ‘(Tujula et al., 2010; Azam et al., 1993; Azam et al., 1998)’ to ‘[26-28]’

line 208: changed ‘(Kirchman et al., 2005; Zhang et al., 2006)’ to ‘[29,30]’

line 210: changed ‘(Kong et al., 2006; Wang et al.,2020a)’ to ‘[31,32]’

line 217: changed ‘(Capone et al., 1997; Liu et al., 1997)’ to ‘[33,34]’

line 219: changed ‘(Woodhouse et al., 2018)’ to ‘[35]’

line 226: changed ‘(Zhang et al., 2018)’ to ‘[36]’

line 227: changed ‘(Wang et al., 2014)’ to ‘[14]’

line 229: changed ‘(Chiellini et al., 2013)’ to ‘[37]’

line 230: changed ‘(Lukwambe et al., 2019)’ to ‘[38]’

line 245: changed ‘(Mohapatra et al., 2020)’ to ‘[39]’

line 247: changed ‘( Kahla et al., 2021)’ to ‘[40]’

line 248: changed ‘(Lee et al., 2016)’ to ‘[41]’

line 280: changed ‘(Walpola et al., 2010)’ to ‘[42]’

line 286: changed ‘(Reed et al., 2011;Vitousek et al.,2013)’ to ‘[43,44]’

line 290: changed ‘(Sawicka et al., 2012)’ to ‘[45]’

line 291: changed ‘(Brauer et al., 2013)’ to ‘[46]’

line 293: changed ‘(Sawicka et al., 2012)’ to ‘[45]

line 302: changed ‘(Seo et al., 2008)’ to ‘[47]’

line 303: changed ‘(Rysgaard and Sloth, 1999)’ to ‘[48]’

line 307: changed ‘(Kelly-Gerreyn et al. 2001)’ to ‘[49]’

line 313: changed ‘(Tiedje et al., 1983; Zumft et al., 1997)’ to ‘[50,51]’

line 318: changed ‘(Ogilvie et al., 1997)’ to ‘[52]’

line 319: changed ‘(Kelly-Gerreyn et al. 2001)’ to ‘[49]’

line 323: changed ‘(Laverman et al., 2007)’ to ‘[53]’

Advice 3. I recommend that the abbreviation be introduced not in the Abstract, but in the text of the manuscript itself at the place of their first mention.

Answer:  We have revised “non-metric multidimensional scaling (NMDS) and Principal co-ordinates analysis (PCoA) showed that Samples exhibited a distinct seasonal distribution, formed three clusters. Redundancy analysis (RDA)” to “Non-metric multidimensional scaling and Principal co-ordinates analysis showed that Samples exhibited a distinct seasonal distribution, formed three clusters. Redundancy analysis ”and changed “Phylogenetic Investigation of Communities by Reconstruction of Unobserved States (PICRUSt) ” to “Phylogenetic Investigation of Communities by Reconstruction of Unobserved States ”.Please see line 21-23 and 27-28.

Advice 4. Be careful with abbreviations, do not forget to decode them at the first mention.

Answer: We have changed “ and RDA. Predict nitrogen metabolism functions in the bacterial community using PICRUSt.” to “ and Redundancy analysis (RDA). Predict nitrogen metabolism functions in the bacterial community using Phylogenetic investigation of communities by reconstruction of unobserved states (PICRUSt).” Please see line 70-74.

Changed “using the QIIME2 (Personalbio Shanghai, China)” to “using the Quantitative Insights Into Microbial Ecology 2 (QIIME2) (Personalbio Shanghai, China)” Please see line 94-95.

Changed “was conducted using SPSS (v2.15.3).” to “was conducted using Statistical Product and Service Solutions (SPSS) (v2.15.3).” Please see line 98-99.

Advice 5. Carefully check the text for typos: italics for Latin taxa designations, capital letters after the comma, absence of spaces between words, etc.

Answer: We have carefully checked the manuscript, those error has been corrected.

Advice 6. Make the Back Matter in accordance with the recommendations of the publisher and in the following order:

Supplementary Materials

Author Contributions: For research articles with several authors, a short paragraph specifying their individual contributions must be provided. The following statements should be used “Conceptualization, X.X. and Y.Y.; methodology, X.X.; software, X.X.; validation, X.X., Y.Y. and Z.Z.; formal analysis, X.X.; investigation, X.X.; resources, X.X.; data curation, X.X.; writing—original draft preparation, X.X.; writing—review and editing, X.X.; visualization, X.X.; supervision, X.X.; project administration, X.X.; funding acquisition, Y.Y. All authors have read and agreed to the published version of the manuscript.” Please turn to the CRediT taxonomy for the term explanation. Authorship must be limited to those who have contributed substantially to the work reported.

Answer: We have revised the format of the article according to the author instructions, Please see manuscript line 349-353.

Credit authorship contribution statement:  Conceptualization, Z.C. and J.L.; methodology, S.L., Z.C. and J.L.; software, M.Z.; formal analysis, M.Z.; investigation, W.D and M.Z.; resources, W.D.; writing—original draft preparation, W.D.; and writing—review and editing, S.L., Z.C. and J.L. funding acquisition, Z.C. and J.L. All authors have read and agreed to the published version of the manuscript.

Advice 7. The references part must be written according to, where the year of publication is put in its correct place and with the correct italics and bold font style.

Answer:

We have revised the format of the partial references according to the requirements of the journal. Please see manuscript line 359-473.

Advice 8. Line 111 “… gradually decreased from spring to winter (7.94-8.71),…” The values in brackets correspond to only one sample out of five, I recommend removing these values from the text, it is clear from the table that there is a gradual decrease in pH.

Answer: We have delate the values from the manuscript. Please see manuscript line 105

Advice 9. Line 113 “…content in the waters; the higher the pH, the gradually decreased…” it is unclear why to refer to the pH value?

Answer:

According to your advice, we find that this is an obvious mistake, and have removed these words from the text. We apologize for the inconvenience caused. Please see manuscript line 107-110.

We have changed “the higher the pH, the gradually decreased the surface salinity from spring to winter, probably because of lower precipitation and river runoff in spring, resulting in saline intrusion, with the high salinity in spring.” to “The surface salinity gradually decreased from spring to winter, it was probably because of lower precipitation and river runoff in spring, resulting in seawater intrusion, then leading to the high salinity in spring.

Advice 10. Figure 3 Add the quadrant designations that are in the text

Answer: According to your advice, we have added the quadrant designations. Please see manuscript line

We have changed “The first axis explained 33.24% of the total variance, whereas the second axis interpreted 14.62%” to “The RDA1 explained 33.24% of the total variance, whereas RDA2 axis interpreted 14.62%

Advice 11. Figure 4 Specify what the indices a, b, c of the sample sites mean, e.g., "a_MTL1"

Answer: The indices a, b, c of the sample site in Figure 4 Specify indicate the spring, summer and winter, respectively. The names of MTL1, HLH2, ZLM3, XD4, YTMT5 mean the sampling sites in the Bailang Estuary.  e.g., "a_MTL1" means sample of site MTL1 in spring. Please see manuscript line 150-151.

Advice 12. Line 213 italics for “Alphaproteobacteria”

Answer: According to your advice, we have changed the “Alphaproteobacteria” to “Alphaproteobacteria”. Please see manuscript line 209.

Advice 13. Line 215-217 “… but Gammaproteobacteria was only correlated with T” negative or positive correlation?

Answer: We have changed “ but Gammaproteobacteria was only correlated with T” to “Gammaproteobacteria was only negative correlated with T ”. Please see manuscript line 212-213.

Advice 14. Line 238-239 “…Cyanobium_PCC-6307, Candidatus_Aquiluna and Synechococcus_CC9902…” incorrect designation of taxa, I believe it means, Cyanobium sp. PCC-6307, Candidatus Aquiluna and Synechococcus sp. CC9902.

Answer:

 According to your advice, we have changed “…Cyanobium_PCC-6307, Candidatus_Aquiluna and Synechococcus_CC9902…” to “Cyanobium sp. PCC- 6307, Candidatus Aquiluna and Synechococcus sp. CC9902.” Please see manuscript line 231-241 and line 332-333.

Round 2

Reviewer 1 Report

Dear Editor,

This version is fine for me and improved enough for publish. 

Author Response

Manuscript No.: jmse-2325703

Manuscript Title: Seasonal and spatial variations of bacterial community structure in the Bailang River estuary

Dear Prof.

Thank you for your advices about the manuscript, those will help us greatly to improve the quality of our articles. We have revised the format of the article according to your advice. Please see Manuscript.

Sincerely yours
